# A tale of osteoarthritis among older adults during the COVID-19 pandemic in Bangladesh: A repeated cross-sectional study

Sabuj Kanti Mistry[1,2,3]*, A. R. M. Mehrab Ali[1], Uday Narayan Yadav[2,4], Rajat Das Gupta[5], Afsana Anwar[6], Saurav Basu[7], Md. Nazmul Huda[1,8,9], Dipak Kumar Mitra[10]

1 ARCED Foundation, Dhaka, Bangladesh, 2 Centre for Primary Health Care and Equity, University of New South Wales, Sydney, Australia, 3 Department of Public Health, Daffodil International University, Dhaka, Bangladesh, 4 National Centre for Epidemiology and Population Health, The Australian National University, Canberra, Australia, 5 Arnold School of Public Health, University of South Carolina, Columbia, South Carolina, United States of America, 6 Health and Nutrition, Social Assistance &, Rehabilitation for the Physically Vulnerable (SARPV), SARPV Complex, Dhaka, Bangladesh, 7 Indian Institute of Public Health–Delhi, New Delhi, India, 8 School of Population Health, The University of New South Wales, Sydney, NSW, Australia, 9 Translational Health Research Institute, Western Sydney University, Campbeltown, NSW, Australia, 10 Department of Public Health, North South University, Dhaka, Bangladesh

* smitra411@gmail.com

**Data Availability Statement:** All relevant data are within the manuscript and its Supporting Information files.

## Abstract

### Background

Due to restrictions in social gatherings imposed due to the COVID-19 pandemic, physical and other daily activities were limited among the older adults. The present study aimed to estimate the change in osteoarthritis prevalence among older adults during the COVID-19 pandemic in Bangladesh.

### Methods

This repeated cross-sectional study was conducted through telephone interviews among older adults aged 60 years and above on two successive occasions (October 2020 and September 2021) during the COVID-19 pandemic in Bangladesh. The prevalence of osteoarthritis was measured by asking the participants if they had osteoarthritis or joint pain problems.

### Results

A total of 2077 participants (1032 in 2020-survey and 1045 in 2021-survey) participated in the study. The prevalence of self-reported joint pains or osteoarthritis significantly increased from 45.3% in 2020 to 54.7% in 2021 ($P = 0.006$), with an increasing odd in the adjusted analysis (aOR 1.27, 95% CI 1.04–1.54). We also found that osteoarthritis prevalence significantly increased among the participants from the Chattogram and Mymensingh divisions, aged 60–69 years, males, married, rural residents, and living with a family. A significant increase was also documented among those who received formal schooling, had a family income of 5000–10000 BDT, resided with a large family, were unemployed or retired, and lived away from a health facility.

**Funding:** The authors received no specific funding for this work

**Competing interests:** The authors have declared that no competing interests exist

## Conclusions

Our study reported a significant increased prevalence of osteoarthritis among older adults from 2020 to 2021 during this pandemic in Bangladesh. This study highlights the need for the development and implementation of initiatives for the screening and management of osteoarthritis through a primary health care approach during any public health emergencies.

## Introduction

The world has been combating an unprecedented COVID-19 pandemic since early 2020 [1]. Globally, over 585 million confirmed cases had been identified with over 64 million deaths, while Bangladesh has seen over 2.0 million confirmed cases with 29,308 deaths as of December 15, 2021 [2]. COVID-19 has created a profound impact on the lives of people with chronic disease due to the diverse nexus of intertwined socio-cultural and biological factors [3]. Given strict lockdown restrictions in many countries during a pandemic, the risk of developing osteoarthritis among older adults could have been mounted because of increased age, higher body mass index (BMI), and reduced physical activity [4].

Osteoarthritis, also refers to as joint pains, is a significant cause of disability and is the most common type of arthritis, affecting 10% of the global population aged 60 years and above [5, 6]. The older population (aged ≥60 years) is projected to double and triple by 2050 and 2100, respectively. It is anticipated that osteoarthritis is primarily related to aging, becoming the leading cause of disability by 2030 [7]. The global prevalence of knee osteoarthritis was estimated to 22.9% in the population aged 40 and above [8]. Osteoarthritis is a prime contributor to the global disability burden and was ranked as the 15th highest cause of years lived with disability (YLDs) in the year 2019 [9], causing a significant financial burden to the health system and societally [6]. Osteoarthritis as comorbidity frequently exists with other non-communicable diseases like heart disease, diabetes, and mental health problems, resulting in further deterioration in the patient's quality of life [9].

According to Osteoarthritis Research Society International (OARSI), physical exercise, physical mobility, and a balanced diet are the core management strategies followed for patients with arthritis [10]. Lockdown measures with increased social isolation, poor social support, the presence of other medical complications, functional disabilities, and loss of physical exercise opportunities can make people more vulnerable to osteoarthritis [11]. The increase of life span in developing countries, including Bangladesh, correlates with an increase in the prevalence of degenerative disorders such as osteoarthritis.

Symptomatic management and promotion of physical exercise are considered the mainstay of osteoarthritis management as it lacks an effective pharmaceutical therapy [5]. However, established guideline to fight against COVID-19 implies "stay at home" and physical distancing" and to implement this strict lockdown had limited the scope of physical exercise for adult people [4]. Home quarantine, and lockdown increasing the probability of ignoring the need of older adults may have complicated the treatment of osteoarthritis. Reduced physical activities also lead to anxiety, and lack of self-motivation may aggravate the disease symptoms [12].

Due to imposed lockdown, cancellation of various events, and social gatherings [13], Bangladeshi older adults have faced challenges in maintaining physical activities and other daily activities. Forms of physical exercise for older adults in Bangladesh are mainly based on social gatherings in tea stalls, parks, larger fields, walking with groups, visiting nearby houses, and doing some chores like grocery shopping, which were limited during these restriction periods.

Restrictions in transportation during pandemic also have limited the access to medical care, physiotherapy sessions, etc., in Bangladesh [13]. These issues could have made older adults more vulnerable to osteoarthritis during the pandemic period in Bangladesh.

Emerging evidence from Bangladesh reported a high prevalence of osteoarthritis and related illnesses among the older population in Bangladesh [14–18]. One study reported that 12.1% of adults aged 60 and above have symptomatic knee osteoarthritis, and the prevalence of osteoarthritis in rural and urban slum and affluent urban communities are 7.5%, 9.2%, and 10.6%, respectively [15]. Findings from the first national survey related to musculoskeletal conditions and related disabilities in Bangladeshi adults show that 30.4% of adults in Bangladesh have musculoskeletal conditions, 7.3% have knee arthritis, and among the adults with musculoskeletal conditions, 24.4% have some form of disabilities [14]. Although a previous study reported an increasing trend in the prevalence of osteoarthritis between 1990 and 2017 [19], we did not find any study that reported the change in osteoarthritis prevalence during the COVID-19 pandemic worldwide, including in Bangladesh. Therefore, to address this knowledge gap, the current study aimed to assess the change in osteoarthritis prevalence during the COVID-19 pandemic and its associated factors among older adults in Bangladesh.

## Materials and methods

### Study design and participants

This repeated cross-sectional study was conducted on two successive occasions, i.e., October 2020 and September 2021, overlapping with the first and second waves of the COVID-19 pandemic in Bangladesh. The study was conducted remotely through telephone interviews by the Aureolin Research, Consultancy and Expertise Development (ARCED) Foundation. The primary challenge for this study was to develop a sampling frame to select participants. Therefore, we utilised our pre-established registry, which included households from all eight administrative divisions of Bangladesh, as a sampling frame [20]. Considering 50% prevalence of osteoarthritis with a 5% margin of error, at the 95% confidence level, 90% power of the test, and 95% response rate, a sample size of 1096 for each survey was calculated. However, during the 2020 survey, 1032 of the approached eligible participants responded to the survey with an overall response rate of approximately 94%. During the 2021 survey, 1045 of approached eligible participants responded to the study with an overall response rate of approximately 95%. Based on the population distribution of older adults by geography in Bangladesh, we adopted a probability proportionate to size (of the eight-division) approach to select older adults aged ≥60 years in each division [21]. Older adults having adverse mental conditions (clinically proved schizophrenia, bipolar mood disorder, dementia/cognitive impairment), a hearing disability, or an inability to communicate were excluded from the study.

### Measures

**Outcome measure.** The outcome of this study was self-reported joint pains or osteoarthritis. The participants were asked, "Do you have osteoarthritis or joint pain problems in the past twelve months preceding the survey?" The response was coded dichotomously as 'Yes' or 'No'.

**Explanatory variables.** Explanatory variables considered in this study were administrative division (Barishal, Chattogram, Dhaka, Mymensingh, Khulna, Rajshahi, Rangpur, Sylhet), age (categorised as 60–69, ≥70), gender (male, female), marital status (married, without a partner), formal schooling (without formal schooling, with formal schooling), family size (≤4, >4), family monthly income (Bangladeshi Taka [BDT]) (<5,000, 5,000–10,000, >10,000), residence

(urban, rural), current occupation (employed, unemployed, retired), living arrangement (living alone, with family), and walking distance to the nearest health centre (<30 min, ≥30 min).

## Data collection tools and techniques

A pre-tested semi-structured questionnaire was used to collect the information via a telephone interview. Data were collected electronically using SurveyCTO mobile app (https://www.surveycto.com/) by trained research assistants, recruited based on previous experience of administering health surveys on the electronic platform. The research assistants were trained extensively before the data collection through Zoom meetings.

The English version of the questionnaire was first translated into Bengali language and then back translated to English by two researchers (SKM, AMA) to ensure the contents' consistency. The questionnaire was then piloted among a small sample (n = 10) of older adults to refine the language in the final version. The tool used in the pilot study did not receive any corrections/suggestions from the participants about the contents developed in the Bengali language.

## Statistical analysis

The distribution of the variables was assessed through descriptive analyses. Given our variables' categorical nature, Chi-square tests were performed to compare differences in the prevalence of osteoarthritis by explanatory variables, with a 5% significance level. We used a binary logistic regression model to estimate the magnitude of association of time (later time compared to an earlier time) with reported osteoarthritis both in the crude model as well as after adjusting for all potential covariates. Crude odds ratio (cOR) and adjusted odds ratio (aOR) with corresponding 95% confidence interval (95% CI) were reported. We checked the multicollinearity in the model using variance inflation factor (VIF), and all VIF values were <5, meaning no multicollinearity. All analyses were performed using the statistical software package Stata (Version 14.0).

## Ethics approval

The study protocol was approved by the Institutional Review Board of the Institute of Health Economics, the University of Dhaka, Bangladesh (Ref: IHE/2020/1037). All procedures performed in studies involving human participants were in accordance with the ethical standards of the institutional and national research committee and with the 1964 Helsinki Declaration and its later amendments or comparable ethical standards. Verbal consent was sought from the participants before administering the survey. Participation was voluntary, and participants did not receive any compensation.

## Patient and public involvement

Patients and/or the public were not involved in developing the research question, study design, conducting study and disseminating results.

## Results

### Characteristics of the participants

Table 1 shows the characteristics of the study participants by survey year. In survey participant coverage, there was a significant difference across geographic areas; for example, the highest coverage was from the Dhaka division in the 2020-Survey, while the highest coverage was from the Khulna division in the 2021-survey. In both surveys, most participants were 60–69

**Table 1. Characteristics of the participants (n = 2077).**

| Characteristics | | Year 2020 | | Year 2021 | | P |
|---|---|---|---|---|---|---|
| | | N | % | n | % | |
| Overall | | 1032 | 100.0 | 1045 | 100.0 | |
| Administrative division | | | | | | |
| | Barishal | 149 | 14.4 | 146 | 14.0 | 0.001 |
| | Chattogram | 137 | 13.3 | 98 | 9.4 | |
| | Dhaka | 210 | 20.4 | 172 | 16.5 | |
| | Mymensingh | 63 | 6.1 | 69 | 6.6 | |
| | Khulna | 158 | 15.3 | 198 | 19.0 | |
| | Rajshahi | 103 | 10.0 | 145 | 13.9 | |
| | Rangpur | 144 | 14.0 | 161 | 15.4 | |
| | Sylhet | 68 | 6.6 | 56 | 5.4 | |
| Age (year) | | | | | | |
| | 60–69 | 803 | 77.8 | 790 | 75.6 | 0.385 |
| | ≥70 | 229 | 22.2 | 255 | 24.4 | |
| Sex | | | | | | |
| | Male | 676 | 65.5 | 620 | 59.3 | 0.004 |
| | Female | 356 | 34.5 | 425 | 40.7 | |
| Marital status | | | | | | |
| | Married | 840 | 81.4 | 799 | 76.5 | 0.006 |
| | Without partner | 192 | 18.6 | 246 | 23.5 | |
| Formal schooling | | | | | | |
| | No formal schooling | 602 | 58.3 | 540 | 51.7 | 0.002 |
| | Having formal schooling | 430 | 41.7 | 505 | 48.3 | |
| Family size | | | | | | |
| | ≤4 | 318 | 30.8 | 347 | 33.2 | 0.243 |
| | >4 | 714 | 69.2 | 698 | 66.8 | |
| Family monthly income (BDT)[1] | | | | | | |
| | <5000 | 145 | 14.1 | 121 | 11.6 | <0.001 |
| | 5000–10000 | 331 | 32.1 | 469 | 44.9 | |
| | >10000 | 556 | 53.9 | 455 | 43.5 | |
| Residence | | | | | | |
| | Urban | 269 | 26.1 | 182 | 17.4 | <0.001 |
| | Rural | 763 | 73.9 | 863 | 82.6 | |
| Current occupation | | | | | | |
| | Employed | 419 | 40.6 | 407 | 39.0 | 0.441 |
| | Unemployed/retired | 613 | 59.4 | 638 | 61.1 | |
| Living arrangement | | | | | | |
| | Living with family | 953 | 92.3 | 992 | 94.9 | 0.016 |
| | Living alone | 79 | 7.7 | 53 | 5.1 | |
| Walking distance to the nearest health centre | | | | | | |
| | <30 minute | 503 | 48.7 | 581 | 55.6 | 0.002 |
| | ≥30 minutes | 529 | 51.3 | 464 | 44.4 | |

[1]1 BDT ~ 0.011 USD.

years old, male, married, without formal schooling, unemployed/retired, and lived with family and in rural areas (Table 1). However, sex, marital status, education, and income significantly

differed across the survey years. Compared with the 2020-survey, a considerably lower proportion of participants in the 2021 survey were males (59% in 2021 vs. 66% in 2020), married (77% vs. 81%), and without formal education (52% vs. 58%). The proportion of participants living with family (95% vs. 93%), in rural areas (83% vs. 74%), and in proximity to a health facility (56% vs. 49%) increased significantly between the survey years (Table 1).

## Changes in the prevalence of osteoarthritis

Table 2 shows the changes in osteoarthritis prevalence over time and its association with participants' characteristics. As seen in Table 2, the osteoarthritis prevalence among the participants significantly increased from 45.3% in 2020 to 54.7% in 2021 ($P = 0.006$).

Table 2 also shows that the osteoarthritis prevalence was significantly increased among the participants who were residing in Chattogram and Mymensingh divisions (47.9% vs 52.1%; 30.8% vs 69.2%, respectively), aged 60–69 years (44.3% vs 55.7%), males (46.5% vs 53.5%), married (47.4% vs 52.6%), rural residents (39.8% vs 60.2%), and living with a family (44.2% vs 55.8%). A significant increase in the prevalence of osteoarthritis was also documented among those who received formal schooling (35.7% vs 64.3%), had a family income of 5000–10000 BDT (36.1% vs 63.9%), were residing with a family having a size of greater than four members (46.0% vs 54.1%), unemployed or retired (43.8%% vs 56.2%), and leaving away from a health facility (47.9% vs 52.1%). Notably, the prevalence of osteoarthritis significantly reduced in urban areas during the two survey years (66.7% vs 33.3%) (Table 2).

After adjusting for all potential covariates, compared with the 2020-survey, the odds of participants' prevalence of osteoarthritis were significantly higher in the 2021-survey (aOR 1.27, 95% CI 1.04–1.54) (Table 3).

## Factors associated with osteoarthritis prevalence

Table 4 shows the factors associated with osteoarthritis in the pooled data. In the unadjusted analysis, age, marital status, formal schooling, family size family monthly income, current employment status, and distance to the nearest health centre was associated with osteoarthritis among the participants at 5% level of significant. However, adjusted analysis shows that osteoarthritis was significantly higher among those aged 70 years and above (aOR: 1.82, 95% CI: 1.46–2.27), who had a family size of more than four members (aOR: 1.26, 95% CI: 1.02–1.55) and who were currently unemployed or retired (aOR: 1.36, 95% CI: 1.11–1.68). On the other hand, osteoarthritis was significantly lower among the participants with a family income of 5000–10000 BDT (aOR: 0.73 95% CI: 0.54–0.99) or >10000 BDT (aOR: 0.63 95% CI: 0.46–0.85) and who was resided more than 30 minutes away from the health centre (aOR: 0.73 95% CI: 0.60–0.89)

## Discussion

To our knowledge, this is the first repeated cross-sectional study that assessed the osteoarthritis prevalence among older adults residing in Bangladesh during the COVID-19 pandemic (between October 2020 and September 2021). Our study found that the prevalence of osteoarthritis significantly increased during the studied period. After adjusting for all potential covariates, multivariable analyses showed an increased odds of osteoarthritis in 2021 compared with 2020. We also found that the osteoarthritis prevalence was significantly increased among the participants who were residing in Chattogram and Mymensingh divisions, aged 60–69 years, males, married ones, rural residents, living with a family, those who received formal schooling, had a family income of 5000–10000 BDT, those having a family size of greater than four

**Table 2. Changes in prevalence of osteoarthritis over time (n = 2077).**

| Characteristics | | Year 2020 | | Year 2021 | | %difference | P |
|---|---|---|---|---|---|---|---|
| | | n | %osteoarthritis | n | % osteoarthritis | | |
| Overall | | 1032 | 45.3 | 1045 | 54.7 | 9.4 | 0.006 |
| Division | | | | | | | |
| | Barishal | 149 | 46.9 | 146 | 53.1 | 6.1 | 0.387 |
| | Chattogram | 137 | 47.9 | 98 | 52.1 | 4.2 | 0.033 |
| | Dhaka | 210 | 53.1 | 172 | 46.9 | -6.2 | 0.633 |
| | Mymensingh | 63 | 30.8 | 69 | 69.2 | 38.5 | 0.012 |
| | Khulna | 158 | 41.9 | 198 | 58.1 | 16.2 | 0.425 |
| | Rajshahi | 103 | 42.9 | 145 | 57.1 | 14.3 | 0.776 |
| | Rangpur | 144 | 43.2 | 161 | 56.8 | 13.6 | 0.400 |
| | Sylhet | 68 | 50.0 | 56 | 50.0 | 0.0 | 0.406 |
| Age (year) | | | | | | | |
| | 60–69 | 803 | 44.3 | 790 | 55.7 | 11.5 | 0.002 |
| | ≥70 | 229 | 47.5 | 255 | 52.5 | 5.0 | 0.944 |
| Sex | | | | | | | |
| | Male | 676 | 46.5 | 620 | 53.5 | 7.0 | 0.012 |
| | Female | 356 | 44.0 | 425 | 56.1 | 12.1 | 0.452 |
| Marital status | | | | | | | |
| | Married | 840 | 47.4 | 799 | 52.6 | 5.3 | 0.034 |
| | Without partners | 192 | 38.8 | 246 | 61.3 | 22.5 | 0.104 |
| Formal schooling | | | | | | | |
| | No formal schooling | 602 | 38.8 | 285 | 61.3 | 22.5 | 0.104 |
| | Having formal schooling | 430 | 35.7 | 208 | 64.3 | 28.6 | 0.000 |
| Family size | | | | | | | |
| | ≤4 | 318 | 43.8 | 347 | 56.3 | 12.5 | 0.181 |
| | >4 | 286 | 46.0 | 698 | 54.1 | 8.1 | 0.013 |
| Family monthly income (BDT)[1] | | | | | | | |
| | <5000 | 145 | 49.1 | 86 | 50.9 | 1.9 | 0.146 |
| | 5000–10000 | 125 | 36.1 | 235 | 63.9 | 27.9 | 0.030 |
| | >10000 | 202 | 52.4 | 172 | 47.7 | -4.7 | 0.274 |
| Residence | | | | | | | |
| | Urban | 269 | 66.7 | 182 | 33.3 | -33.3 | 0.044 |
| | Rural | 763 | 39.8 | 863 | 60.2 | 20.4 | 0.000 |
| Current occupation | | | | | | | |
| | Employed | 419 | 48.6 | 407 | 51.4 | 2.8 | 0.469 |
| | Unemployed/retired | 613 | 43.8 | 638 | 56.2 | 12.5 | 0.005 |
| Living arrangement | | | | | | | |
| | Living with family | 387 | 44.2 | 392 | 55.8 | 11.6 | 0.003 |
| | Living alone | 29 | 60.9 | 24 | 39.1 | -21.7 | 0.861 |
| Distance from nearest health centre | | | | | | | |
| | <30 minute | 503 | 43.4 | 581 | 56.6 | 13.2 | 0.141 |
| | ≥30 minutes | 529 | 47.9 | 464 | 52.1 | 4.2 | 0.031 |

[1]1 BDT ~ 0.011 USD.

members, unemployed or retired, and living away from a health facility. On the other hand, a significant reduction in the osteoarthritis prevalence was observed in the urban participants.

**Table 3. Changes in prevalence of osteoarthritis after adjusting for potential covariates (n = 2077).**

| Characteristics | | cOR[1] | 95% CI | P | aOR[2] | 95% CI | P |
|---|---|---|---|---|---|---|---|
| Self-reported osteoarthritis | | | | | | | |
| | 2020 Survey | Ref | | | Ref | | |
| | 2021 Survey | 1.30 | 1.08–1.56 | 0.006 | 1.27 | 1.04–1.54 | 0.016 |

[1]Crude Odds Ratio;

[2]Adjusted Odds Ratio.

*Model adjusted for all the covariates reported in Table 2.

The increase in the prevalence of osteoarthritis among older adults might result from increased Body Mass Index (BMI) due to decreased normal physical activities during the COVID-19 pandemic [22]. Throughout the pandemic, many older adults were confined in their homes because of the lockdowns and restrictions imposed by the government to prevent COVID-19 infections [13]. It is well-established that lack of physical inactivity increases the risk of overweight and obesity in older people, a well-established risk factor for osteoarthritis [23, 24].

Our study found a higher prevalence of osteoarthritis in rural participants than in urban ones. This suggests the adoption of sedentary lifestyles secondary to pandemic-related loss of jobs [4]. A nationwide study by BRAC (Building Resources Across Communities) reported that at least one earning member of nearly 61% of families with internal migration lost a job during the pandemic [25]. For a similar reason, unemployed or retired participants were at risk of developing osteoarthritis and the prevalence increased significantly among them. Our argument is supported by a study conducted on South Korean adults aged ≥ 50 years, that found that sedentary lifestyles were associated with increased odds of chronic knee pain and knee osteoarthritis [26].

The current study showed that participants living with a family or residing with a family with a size greater than four members significantly increased in the prevalence of osteoarthritis. The older individuals in these families are more likely to be physically inactive due to assisting members, which increases the risk of aggravating osteoarthritis. Higher family income and education were also associated with a significant increase in prevalence. In lower middle-income countries like Bangladesh, people with a higher wealth index and higher educational attainment frequently adopt sedentary lifestyles in the name of modernization, a known risk factor for triggering osteoarthritis and reducing their associated quality of life [27].

We also observed a significant increase in the prevalence of osteoarthritis among the individuals who resided away from a health facility. In general, individuals suffering from osteoarthritis also suffer from other comorbidities, including obesity, diabetes, hypertension, and cardiovascular diseases [4]. A proximal residence near the health system may improve the health-seeking behaviours for the condition and adopt ameliorative and preventive measures. The significant increase in Chattogram and Mymensingh division requires further exploration to understand the complex socio-ecological or geographical features which lead to this change.

Zahid Al Quadir (2020) estimated the prevalence of knee osteoarthritis in Bangladesh before the COVID-19 pandemic to be 7.3% (95% CI: 6.1%–8.5%) [14], with the older population being the most vulnerable group. However, our study indicated that more than half of the respondents suffered joint pains suggestive of osteoarthritis, probably due to a sedentary lifestyle that they adopted during the ongoing pandemic.

Osteoarthritis is a degenerative condition linked to aging process, and the older population in Bangladesh is projected to rise by 40% by 2050, a factor that is expected to increase the chronic

**Table 4. Factors associated with osteoarthritis prevalence in the pooled data (n = 2077).**

| Characteristics | | cOR[1] | 95% CI | P | aOR[2] | 95% CI | P |
|---|---|---|---|---|---|---|---|
| Age (years) | | | | | | | |
| | 60–69 | Ref | | | Ref | | |
| | ≥70 | 2.07 | 1.68–2.56 | **<0.001** | 1.82 | 1.46–2.27 | **<0.001** |
| Marital status | | | | | | | |
| | Married | Ref | | | Ref | | |
| | Without partners | 1.26 | 1.01–1.58 | **0.038** | 0.95 | 0.75–1.21 | 0.669 |
| Formal schooling | | | | | | | |
| | No formal schooling | Ref | | | Ref | | |
| | Having formal schooling | 0.81 | 0.68–0.98 | **0.031** | 0.85 | 0.70–1.03 | 0.102 |
| Family size | | | | | | | |
| | ≤4 | Ref | | | Ref | | |
| | >4 | 1.27 | 1.04–1.56 | **0.018** | 1.26 | 1.02–1.55 | **0.033** |
| Family monthly income (BDT)[3] | | | | | | | |
| | <5000 | Ref | | | Ref | | |
| | 5000–10000 | 0.76 | 0.57–1.02 | 0.066 | 0.73 | 0.54–0.99 | **0.043** |
| | >10000 | 0.63 | 0.48–0.83 | **0.001** | 0.63 | 0.46–0.85 | **0.002** |
| Residence | | | | | | | |
| | Urban | Ref | | | Ref | | |
| | Rural | 1.11 | 0.89–1.39 | 0.355 | 1.06 | 0.83–1.35 | 0.643 |
| Current occupation | | | | | | | |
| | Employed | Ref | | | Ref | | |
| | Unemployed/retired | 1.63 | 1.34–1.97 | **<0.001** | 1.36 | 1.11–1.68 | **0.003** |
| Living arrangement | | | | | | | |
| | Living with family | Ref | | | Ref | | |
| | Living alone | 1.12 | 0.78–1.63 | 0.535 | 0.99 | 0.66–1.48 | 0.962 |
| Distance from nearest health centre | | | | | | | |
| | <30 minute | Ref | | | Ref | | |
| | ≥30 minutes | 0.73 | 0.61–0.88 | **0.001** | 0.73 | 0.60–0.89 | **0.002** |

[1]Crude Odds Ratio;

[2]Adjusted Odds Ratio;

[3]1 BDT ~ 0.011 USD.

disease burden. Although the public health programs are well placed in Bangladesh, care components focusing on older adults are very limited, with primary care facilities often equipped with limited trained physicians required for screening and management of osteoarthritis for this vulnerable population [28]. Existing chronic disease care initiatives should incorporate osteoarthritis prevention, screening, and safe symptomatic management elements, required to manage pain and disability resulting from this condition. Physical literacy that involves building skills, knowledge and behaviors for active lives and physical activity that can be done in indoor settings during the pandemic period should be offered to the older population to prevent osteoarthritis. Future mixed-method studies are also suggested to precisely understand the mechanism of increasing the prevalence of osteoarthritis in specific subgroups of population.

## Strength and limitations

To our knowledge, this is the first study reporting the change in the osteoarthritis prevalence and factors associated with osteoarthritis among the older population during the COVID-19

pandemic in Bangladesh. However, the study was subjected to several limitations which warrant discussion. First, the collected information was self-reported. Osteoarthritis was not objectively measured through radiographic evidence, allowing a chance of misclassification. However, we assume that this misclassification was non-differential in direction. Second, we could not collect data on height and weight. As a result, we could not examine the association between changes in BMI with the change in the prevalence of osteoarthritis. Third, the information on the health-seeking behaviour of the participants was also lacking. Fourth, we could not follow the same person over the time i.e., panel, but followed a repeated cross-sectional design. Thus, the change suggests a trend observed in a population and does not indicate changes experienced at the individual level. Fifth, our study is limited to quantitative analysis and points to potential factors, but a qualitative study may provide insights into the underlying reasons for the change. These limitations highlight the need for further studies with longitudinal analysis and a mixed-method approach among the older population in Bangladesh during this pandemic.

## Conclusion

The present study reported an increased prevalence of osteoarthritis and its associates among the older population during COVID-19 pandemic in Bangladesh. The findings from this study highlights the need of screening, prevention, and management of osteoarthritis within a primary care approach. Our findings could also guide policymakers and public health practitioners to develop initiatives that could create supportive environment for people with osteoarthritis to access health services and perform daily activities including physical activity required to maintain a good quality of life during this pandemic.

## Supporting information

**S1 Data. Data file of the study.**
(DTA)

## Acknowledgments

We acknowledge the role of Sadia Sumaia Chowdhury, Programme Manager, ARCED Foundation and Md. Zahirul Islam, Project Associate, ARCED Foundation, for their support in data collection for the study.

## Author Contributions

**Conceptualization:** Sabuj Kanti Mistry.

**Formal analysis:** Sabuj Kanti Mistry.

**Investigation:** Sabuj Kanti Mistry.

**Methodology:** Sabuj Kanti Mistry.

**Software:** Sabuj Kanti Mistry.

**Supervision:** Sabuj Kanti Mistry.

**Writing – original draft:** Sabuj Kanti Mistry, A. R. M. Mehrab Ali, Uday Narayan Yadav, Rajat Das Gupta, Afsana Anwar, Saurav Basu, Md. Nazmul Huda.

**Writing – review & editing:** Dipak Kumar Mitra.

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
