## [Decision Letter · Decision Letter 0]

4 Aug 2022

PONE-D-22-12111A tale of osteoarthritis among older adults during the COVID-19 pandemic in Bangladesh: A repeated cross-sectional studyPLOS ONE

Dear Dr. Mistry,

Thank you for submitting your manuscript to PLOS ONE. After careful consideration, we feel that it has merit but does not fully meet PLOS ONE’s publication criteria as it currently stands. Therefore, we invite you to submit a revised version of the manuscript that addresses the points raised during the review process

We look forward to receiving your revised manuscript.

Kind regards,

Md. Tanvir Hossain

Academic Editor

PLOS ONE

Journal Requirements:

"No"

"No"

 This information should be included in your cover letter; we will change the online submission form on your behalf."

Reviewers' comments:

Reviewer's Responses to Questions

**Comments to the Author**

1. Is the manuscript technically sound, and do the data support the conclusions?

Reviewer #1: Yes

Reviewer #2: Yes

2. Has the statistical analysis been performed appropriately and rigorously? 

Reviewer #1: Yes

Reviewer #2: Yes

3. Have the authors made all data underlying the findings in their manuscript fully available?

Reviewer #1: Yes

Reviewer #2: No

4. Is the manuscript presented in an intelligible fashion and written in standard English?

Reviewer #1: Yes

Reviewer #2: No

5. Review Comments to the Author

Reviewer #1: Thanks to the authors for addressing an important issue and I am happy reading the paper. It is a well written manuscript and easy to understand the rationality.

The rationality is clear and well organized manuscript. I am happy reading the paper.

I don’t have much comments just quarries is the outcome variable we self reported?

What is the justification using self reported disease for a medical research?

Please re edit the conclusion of the abstract it overtly written please be specific in writing the conclusion.

Reviewer #2: The manuscript " A tale of osteoarthritis among older adults during the COVID-19 pandemic in Bangladesh: A repeated cross-sectional study is interesting. However, after reading the manuscript, I have certain reservations.

1. Authors are advised to correct their grammatical mistakes throughout the whole manuscript.

2. Incomplete sentences Under introduction lines 107-108.

3. Research gaps and objectives should be clearer.

4. The inclusion criteria under the section ‘Study design and participants’ adopted age should be >= 60 to match the analysis if I am not wrong.

5. In table 1, the format of the >= symbol should be the same for all indicators. For age, you used >= 70 but for family size you used ≤4.

6. Table 2 and the data presentation is not in the proper manner. Authors are advised to correct the table's format.

7. Authors are advised to show the results of binary logistic regression for all covariates, both for cOR and aOR, by creating a separate table.

6. PLOS authors have the option to publish the peer review history of their article (what does this mean?). If published, this will include your full peer review and any attached files.

Reviewer #1: No

Reviewer #2: No

---

## [Decision Letter · Decision Letter 1]

6 Sep 2022

A tale of osteoarthritis among older adults during the COVID-19 pandemic in Bangladesh: A repeated cross-sectional study

PONE-D-22-12111R1

Dear Dr. Mistry,

We’re pleased to inform you that your manuscript has been judged scientifically suitable for publication and will be formally accepted for publication once it meets all outstanding technical requirements.

Kind regards,

Md. Tanvir Hossain

Academic Editor

PLOS ONE

Reviewers' comments:

Reviewer's Responses to Questions

**Comments to the Author**

1. If the authors have adequately addressed your comments raised in a previous round of review and you feel that this manuscript is now acceptable for publication, you may indicate that here to bypass the “Comments to the Author” section, enter your conflict of interest statement in the “Confidential to Editor” section, and submit your "Accept" recommendation.

Reviewer #1: All comments have been addressed

Reviewer #2: All comments have been addressed

2. Is the manuscript technically sound, and do the data support the conclusions?

Reviewer #1: Partly

Reviewer #2: Yes

3. Has the statistical analysis been performed appropriately and rigorously? 

Reviewer #1: Yes

Reviewer #2: Yes

4. Have the authors made all data underlying the findings in their manuscript fully available?

Reviewer #1: Yes

Reviewer #2: Yes

5. Is the manuscript presented in an intelligible fashion and written in standard English?

Reviewer #1: Yes

Reviewer #2: Yes

6. Review Comments to the Author

Reviewer #1: (No Response)

Reviewer #2: (No Response)

7. PLOS authors have the option to publish the peer review history of their article (what does this mean?). If published, this will include your full peer review and any attached files.

Reviewer #1: No

Reviewer #2: No

---

## [Editor Report · Acceptance letter]

9 Sep 2022

PONE-D-22-12111R1 

A tale of osteoarthritis among older adults during the COVID-19 pandemic in Bangladesh: A repeated cross-sectional study 

Dear Dr. Mistry:

I'm pleased to inform you that your manuscript has been deemed suitable for publication in PLOS ONE. Congratulations! Your manuscript is now with our production department. 

Kind regards, 

on behalf of

Dr. Md. Tanvir Hossain 

Academic Editor

PLOS ONE